



**Fluorescence lidar observations of wildfire smoke inside cirrus: A contribution to smoke-**
**cirrus - interaction research**
Igor Veselovskii[1], Qiaoyun Hu[2], Albert Ansmann[3], Philippe Goloub[2], Thierry Podvin[2], Mikhail
Korenskiy[1]
[1]*Prokhorov General Physics Institute of the Russian Academy of Sciences, Moscow, Russia.*
[2]*Univ. Lille, CNRS, UMR 8518 - LOA - Laboratoire d'Optique Atmosphérique, Lille, 59000,*
*France*
[3]*Leibniz Institute for Tropospheric Research, Leipzig, Germany*
**Correspondence**: Igor Veselovskii (iveselov@hotmail.com)
**Abstract**
A remote sensing method, based on fluorescence lidar measurements, that allows to detect and to
quantify the smoke content in upper troposphere and lower stratosphere (UTLS) is presented.
The unique point of this approach is that, smoke and cirrus properties are observed in the same
air volume simultaneously. In the article, we provide results of fluorescence and
multiwavelength Mie-Raman lidar measurements performed at ATOLL observatory from
Laboratoire d'Optique Atmosphérique, University of Lille, during strong smoke episodes in the
summer and autumn seasons of 2020. The aerosol fluorescence was induced by 355 nm laser
radiation and the fluorescence backscattering was measured in a single spectral channel, centered
at 466 nm of 44 nm width. To estimate smoke properties, such as number, surface area and
volume concentration, the conversion factors, which link the fluorescence backscattering and the
smoke microphysical properties, are derived from the synergy of multiwavelength Mie-Raman
and fluorescence lidar observations. Based on two case studies, we demonstrate that the
fluorescence lidar technique provides possibility to estimate the smoke surface area
concentration within freshly formed cirrus layers. This value was used in smoke INP
parameterization scheme to predict ice crystal number concentrations in cirrus generation cells.
**Introduction**

Aerosol particles in the upper tropospheric and lower stratospheric (UTLS) height regime

play an important role in processes of heterogeneous ice formation, however our current
understanding of these processes is insufficient for a trustworthy implementation in numerical



weather and climate prediction models. The ability of aerosol particles to act as ice nucleating
particles (INP) depends on meteorological factors such as temperature and ice supersaturation (as
a function of vertical velocity) as well as on the aerosol type in the layer in which cirrus
developed (Kanji et al., 2017). Heterogeneous ice nucleation initiated by insoluble inorganic
materials such as mineral dust has been studied since a long time (e.g., DeMott et al 2010, 2015;
Hoose and Möhler, 2012; Murray et al., 2012; Boose et al., 2016; Schrod et al., 2017; Ansmann
et al. 2019b), while the potential of omnipresent organic particles, especially of frequently
occurring aged, long-range-transported wildfire smoke particles, to act as INP is less well
explored and thus not well understood (Knopf et al., 2018). Wildfire smoke can reach the lower
stratosphere via pyro-cumulonimbus (pyroCb) convection (Fromm et al., 2010; Peterson et al.,
2018, 2021; Hu et al., 2019; Khaykin et al., 2020) or via self-lifting processes (Boers et al.,
2010,Ohneiser et al., 2021). It is widely assumed that the ability of smoke particles to serve as
INP mainly depends on the organic material (OM) in the shell of the coated smoke particles
(Knopf et al., 2018), but may also depend on mineral components in the smoke particles (Jahl et
al., 2021).  The ice nucleation efficiency may increase with increasing duration of the long-range
transport as Jahl et al. (2021) suggested. Disregarding the progress made in this atmospheric
research field during the last years, the link between ice nucleation efficiency and the smoke
particle chemical and morphological properties is still largely unresolved (China et al., 2017;
Knopf et al., 2018).
To contribute to the field of smoke-cirrus-interaction research, we present a remote
sensing method that allows us simultaneously to detect and quantify the smoke amount inside of
cirrus layers together with cirrus properties and to provide INP estimates in regions close to
cloud top where ice formation usually begins. The unique point of our approach is that, for the
first time, smoke and cirrus properties are observed in the same air volume simultaneously.
Recently, a first attempt (closure study) was performed to investigate the smoke impact on High
Arctic cirrus formation (Engelmann et al. 2021). However, the aerosol measurements had to be
performed outside the clouds layers, and then the assumption was needed that the found aerosol
(and estimated INP) concentration levels also hold inside the cirrus layers. Now, we are able to
directly determine INP-relevant smoke parameters inside the cirrus layer during ice nucleation
events. This also offers the opportunity to illuminate whether an INP reservoir can be depleted in





cirrus evolution processes or not. Furthermore, this new lidar detection method permits a clear
discrimination between, e.g., smoke and mineral dust INPs.
Multiwavelength Mie-Raman lidars or High Spectral Resolution lidars (HSRL) are
favorable instruments to profile the physical properties of aerosol particles in troposphere. In
particular, the inversion of so-called 3β+2α lidar observations, based on the measurement of
height profiles of three aerosol backscatter coefficients at 355, 532, and 1064 nm and two
extinction coefficients at 355 and 532 nm, allows us to estimate smoke microphysical properties
(Müller et al., 1999, 2005; Veselovskii et al., 2002, 2015). However, the aerosol content in
UTLS height range can be low, so that particle extinction coefficients cannot be determined with
sufficient accuracy and are thus not available in the lidar inversion data analysis. To resolve this
issue Ansmann et al. (2019a, 2021) used the synergy of polarization lidar measurements and
Aerosol Robotic Network (AERONET) sunphotometer observations (Holben et al., 1998) to
derived conversion factors (to convert backscatter coefficients into microphysical particle
properties) and to estimate INP concentrations for dust and smoke aerosols with the retrieved
aerosol surface area concentration as aerosol input.
Dust particles are very efficient ice nuclei in contrast to wildfire smoke particles. In this
context, the question arises: How can we unambiguously discriminate smoke from dust
particles? This is realized by integrating a fluorescence channel into a multiwavelength aerosol
lidar (Reichardt et al., 2017; Veselovskii et al., 2020; 2021). The fluorescence capacity of smoke
(ratio of fluorescence backscattering to the overall aerosol backscattering), significantly exceeds
corresponding values for other types of aerosol, such as dust or anthropogenic particles
(Veselovskii et al., 2020; 2021), and thus allows us to discriminate smoke from other aerosol
types. The fluorescence technique provides therefore the unique opportunity to monitor ice
formation in well identified wildfire smoke layers, and thus to create a good basis for long-term
investigations of smoke cirrus interaction.
In this article, we present results of fluorescence and multiwavelength Mie-Raman lidar
measurements performed at the ATOLL (ATmospheric Observation at liLLe) at Laboratoire
d'Optique Atmosphérique, University of Lille, during strong smoke episodes in the summer and
autumn seasons of 2020. The results demonstrate that the fluorescence lidar is capable to monitor
the smoke in the UTLS height range and inside the cirrus clouds formed at or below the
tropopause. We start with a brief description of the experimental setup in Sect.2. In the first part





of the result section (Sect.3.1 and 3.2), it is explained how smoke optical properties can be
quantified by using fluorescence backscattering information and how we can estimate smoke
microphysical properties (volume, surface area, and number concentration) from measured
fluorescence backscatter coefficients. In this approach, multiwavelength Mie-Raman aerosol
lidar observations are used in addition. Values of the smoke particle surface area concentration
are then the aerosol input in the smoke INP estimation. A case study is discussed in Sect.3.2.
Two case studies are then presented in Sect.3.3 to demonstrate the capability of a fluorescence
lidar to monitor ice formation in extended smoke layers and to provide detailed information on
aerosol microphysical properties and smoke-relate INP concentration levels.

**2. Experimental setup**
The multiwavelength Mie-Raman lidar LILAS (LIlle Lidar AtmosphereS) is based on a
tripled Nd:YAG laser with a 20 Hz repetition rate and pulse energy of 70 mJ at 355 nm.
Backscattered light is collected by a 40 cm aperture Newtonian telescope and the lidar signals
are digitized with Licel transient recorders of 7.5 m range resolution, allowing simultaneous
detection in the analog and photon counting mode. The system is designed for simultaneous
detection of elastic and Raman backscattering, allowing the so called $3\beta+2\alpha+3\delta$ data
configuration, including three particle backscattering ($\beta_{355}$, $\beta_{532}$, $\beta_{1064}$), two extinction ($\alpha_{355}$, $\alpha_{532}$)
coefficients along with three particle depolarization ratios ($\delta_{355}$, $\delta_{532}$, $\delta_{1064}$). The particle
depolarization ratio, determined as a ratio of cross- and co-polarized components of the particle
backscattering coefficient, was calculated and calibrated in the same way as described in
Freudenthaler et al. (2009). The aerosol extinction and backscattering coefficients at 355 and 532
nm were calculated from Mie-Raman observations (Ansmann et al., 1992), while $\beta_{1064}$ was
derived by the Klett method (Fernald, 1984; Klett, 1985). Additional information about
atmospheric parameters was available from radiosonde measurements performed at
Herstmonceux (UK) and Beauvechain (Belgium) stations, located 160 km and 80 km away
from the observation site respectively.
The lidar system is also capable to perform aerosol fluorescence measurements. A part of
the fluorescence spectrum is selected by a wideband interference filter of 44 nm width centered
at 466 nm (Veselovskii et al., 2020; 2021). The strong sunlight background at daytime restricts
the fluorescence observations to nighttime hours. To characterize the fluorescence properties of



aerosol, the fluorescence backscattering coefficient $\beta_F$ is calculated from the ratio of
fluorescence and nitrogen Raman backscatters, as described in Veselovskii et al. (2020). This
approach allows to evaluate the absolute values of $\beta_F$, if the relative sensitivity of the channels is
calibrated and the nitrogen Raman scattering differential cross section $\sigma_R$ is known. In our
research we used $\sigma_R=2.744*10^{-30}$ cm$^2$sr$^{-1}$ at 355 nm from Venable et al. (2011). All $\beta_F$ profiles
presented in this work were smoothed with the Savitzky – Golay method, using second order
polynomial with 21 points in the window. The efficiency of fluorescence backscattering with
respect to elastic backscattering $\beta_{532}$ is characterized by the fluorescence capacity $G_F = \dfrac{\beta_F}{\beta_{532}}$ .

For most of atmospheric particles $\beta_F$ is proportional to the volume of dry matter, while

dependence of $\beta_{532}$ on particle size is more complicated. As a result, $G_F$ depends not only on
aerosol type, but also on particle size and the relative humidity RH. We recall also, that only a
part of the fluorescence spectra was selected by the interference filter in the receiver, so provided
values of $\beta_F$ and $G_F$ are specific for the filter used. Analyzing the fluorescence measurements we
should keep in mind, that the sensitivity of this technique can be limited by the fluorescence of
optics in the lidar receiver. The minimal value of $G_F$, which we measured during observation in
cloudy conditions in the lower troposphere was about $2\times10^{-8}$. Thus, at least, in the measurements
with $G_F$ above this value, the contribution of optics fluorescence can be ignored.

**3. Results of the measurements**
*3.1. Observation of smoke particles in UTLS*

Smoke particles produced by intensive fires and transported across the Atlantic are

regularly observed in the UTLS height range over Europe (Müller et al., 2005; Hu et al., 2019;
Baars et al., 2019, 2021). One of such events, observed over Lille in the night of 4-5 November
2020, is shown in Fig.1. The figure provides height – time displays of the range corrected lidar
signal and the volume depolarization ratio at 1064 nm together with the fluorescence
backscattering coefficient. A narrow smoke layer occurred in the upper troposphere in the period
from 23:00 – 06:00 UTC. The smoke was detected at heights above 12 km after midnight. The
particles caused a low volume depolarization ratio (<5%) at 1064 nm and strong fluorescence
backscattering ($\beta_F>1.2\times10^{-4}$ Mm$^{-1}$sr$^{-1}$). The backward trajectory analysis indicated that the





aerosol layer was transported over the Atlantic and contained products of North American wild
fires.

Vertical profiles of aerosol $\beta_{532}$ and fluorescence $\beta_F$ backscattering coefficients for the

period from 02:00 - 05:30 UTC are shown in Fig.2a. The fluorescence capacity $G_F$ in the center
of smoke layer (not shown) was about $4.5\times10^{-4}$. The depolarization ratio of aged smoke in the
UTLS height range usually shows a strong spectral dependence (Haarig et al., 2018; Hu et al.,
2019). For the case presented in Fig.2a the particle depolarization ratio in the center of the smoke
layer decreased from 16±4% at 355 nm ($\delta_{355}$) to 4±1% at 1064 ($\delta_{1064}$). The tropopause height $H_{tr}$
was at about 13000 m, thus the main part of the smoke layer was below the tropopause. By the
end of day the smoke layer became weaker ($\beta_F<0.3\times10^{-4}$ Mm$^{-1}$sr$^{-1}$) and ascended up to 14500 m,
which is above the tropopause. Corresponding vertical profiles of $\beta_{532}$ and $\beta_F$ are shown in
Fig.2b. The fluorescence capacity in the center of the layer is about $4.5\times10^{-4}$, which is close to
the value observed during 02:00 - 05:30 UTC period.

An important advantages of the fluorescence lidar technique is the ability to monitor

smoke particles inside cirrus clouds. The results of smoke observations in the presence of ice
clouds are shown in Fig.3. Cirrus clouds occurred during the whole night in the height range
from 6.0 km – 10.0 km. To quantify the fluorescence backscattering inside the cloud (which was
rather weak in this case), the lidar signals were averaged over the full 18:00 – 06:00 UTC time
interval in Fig.3a. The fluorescence  backscatter coefficient shown in Fig.3c decreased from
$\beta_F=0.015\times10^{-4}$ Mm$^{-1}$sr$^{-1}$ at 5000 m (near the cloud base) to a minimum value of $0.01\times10^{-4}$ Mm$^{-}$
$^1$sr$^{-1}$ at 7000 m inside the cirrus layer. Above the tropopause the fluorescence backscattering
increased strongly and reached the maximum (about $0.3\times10^{-4}$ Mm$^{-1}$sr$^{-1}$) in 11000 m -13000 m
height.

The analysis of fluorescence measurements performed during strong smoke episodes in

the summer and autumn of 2020, when smoke layers from North American fires frequently
reached Europe, demonstrates that the fluorescence capacity varied within the range of $2.8\times10^{-4}$
to $4.5\times10^{-4}$. The variations are a function of smoke composition, relative humidity and particle
size. However, in the upper troposphere, where relative humidity is low, $G_F$ was normally close
to $4.5\times10^{-4}$. This relatively low range of $G_F$ variations allows the estimation of the backscattering
coefficient attributed to the smoke particles from fluorescence measurements as:





$$\beta_{532}^s = \frac{\beta_F}{G_F}. \tag{1}$$

Fig.3d shows the smoke backscattering coefficient $\beta_{532}^s$, calculated from $\beta_F$ for

$G_F = 4.5 \times 10^{-4}$, together with $\beta_{532}$. The dynamical range of $\beta_{532}$ variations is high. To make smoke
backscattering visible above $H_{Tr}$, $\beta_{532}$ is plotted in expanded scale in Fig.3d. The $\beta_{532}^s$ values,
though being strongly oscillating above the tropopause, match the $\beta_{532}$ indicating that the smoke
contribution to backscattering was predominant.

**3.2. Estimation of smoke particles content based on fluorescence measurements**

The possibility to detect fluorescence backscattering inside the cirrus clouds reveals also

the opportunity for a quantitative characterization of the smoke content. This can be realized by a
synergistic use of fluorescence and multiwavelength Mie – Raman lidar observations. For the
smoke layers with sufficient optical depth, the number $N$, surface area $S$ and volume $V$
concentrations can be evaluated, by inverting the $3\beta + 2\alpha$ observations consisting of three
backscatter coefficients (355, 532, 1064 nm) and two extinction coefficients (355, 532 nm)
(Müller et al., 1999; Veselovskii et al., 2002). The conversion factors $C_N$, $C_S$, $C_V$, introduced as
$$C_N = \frac{N}{\beta_F}, \ C_S = \frac{S}{\beta_F}, \ C_V = \frac{V}{\beta_F}, \tag{3}$$

allow the estimation of $N$, $S$, and $V$ from fluorescence backscattering.

On 23-24 June 2020, a strong smoke layer was observed in $4500 - 5500$ m height during

the whole night (Fig.4). The vertical profiles of the aerosol backscattering and extinction
coefficients ($3\beta + 2\alpha$) are shown in Fig.5a, while the particle depolarization ratios $\delta_{355}$, $\delta_{532}$, $\delta_{1064}$
and the lidar ratios at 355 nm and 532 nm ($LR_{355}$, $LR_{532}$) are presented in Fig.5b. The
depolarization ratio decreases with wavelength from $9 \pm 1.5\%$ at 355 nm to $1.5 \pm 0.3\%$ at 1064 nm
and the lidar ratio at 532 nm significantly exceeds corresponding value at 355 nm ($80 \pm 12$ sr and
$50 \pm 7.5$ sr respectively), which is typical for aged smoke (Müller et al., 2005). The
multiwavelength observations were inverted to determine the particle effective radius $r_{eff}$,
number, surface area and volume concentrations for seven height bins inside the smoke layer.
The effective radius $r_{eff}$ in Fig.5c increases through the layer from 0.15 μm to 0.2 μm
simultaneously with the increase of the fluorescence capacity $G_F$ from $2.8 \times 10^{-4}$ to $3.6 \times 10^{-4}$.



Retrieved values of $N$, $S$, $V$ were used for calculation of conversion factors (Eq. 3) for each
height bin. In the center of the smoke layer (at 4.9 km) the factors are: $C_N = 88 \times 10^4 \, \frac{cm^{-3}}{Mm^{-1}sr^{-1}}$,
$C_S = 35 \times 10^4 \, \frac{\mu m^2 cm^{-3}}{Mm^{-1}sr^{-1}}$, and $C_V = 2.4 \times 10^4 \, \frac{\mu m^3 cm^{-3}}{Mm^{-1}sr^{-1}}$. Thus, when $\beta_F$ is given in $Mm^{-1}sr^{-1}$, the
calculated values of $N$, $S$, and $V$ are given in $cm^{-3}$, $\mu m^2 cm^{-3}$ and $\mu m^3 cm^{-3}$ respectively.
Fluorescence backscattering is proportional to the particle volume concentration, so $C_V$ is not
sensitive to the effective radius variation. The conversion factors $C_N$ and $C_S$, on the contrary,
depend on the particle size. Fig.5d shows the profiles of $N$, $S$, $V$ obtained by inversion of 3β+2α
observations (symbols) together with corresponding values ($N_F$, $S_F$, $V_F$) obtained from $\beta_F$, using
the mean conversion factors for seven height bins considered. The volume concentrations $V$ and
$V_F$ agree well for all seven height bins. For the surface area concentrations the agreement is still
good, but for $N$ and $N_F$ the difference is up to 30%. We need to emphasize, that the conversion
factors presented are specific for our lidar system (for the interference filter installed in
fluorescence channel). It is worthwhile to mention that the ratio $V/\alpha_{532}$ of the volume
concentration $V$ in Fig.5d to the extinction coefficient α at 532 nm in Fig.5a, as well as the ratio
$S/\alpha_{532}$, are very close to respective extinction-to-volume and extinction-to-surface-area-
concentration conversion factors presented for aged wildfire smoke by Ansmann et al. (2021).

The conversion factors depend on the smoke composition. To estimate the variation range

of $C_N$, $C_S$, $C_V$, several smoke episodes were analyzed and corresponding results are presented in
Table 1. The table provides the fluorescence capacity $G_F$ and the conversion factors at the
heights, where 3β+2α data could be calculated. Mean values of $C_N$, $C_S$, $C_V$ derived for these
episodes and corresponding standard deviations are:
$C_N = (61 \pm 32) \times 10^4 \frac{cm^{-3}}{Mm^{-1}sr^{-1}}$;  $C_S = (28 \pm 6.4) \times 10^4 \frac{\mu m^2 cm^{-3}}{Mm^{-1}sr^{-1}}$;  $C_V = (2.2 \pm 0.2) \times 10^4 \frac{\mu m^3 cm^{-3}}{Mm^{-1}sr^{-1}}$    (4)
Thus, the expected uncertainties in the $N$, $S$ and $V$ estimations from fluorescence measurements
are of 50%, 25%, and 10% respectively, which is comparable with uncertainty of inversion of
3β+2α observations (Veselovskii et al., 2002; Pérez-Ramírez et al., 2013). Table 1 shows also
the volume and surface area concentrations of the smoke particles obtained from the inversion of
*3β+2α* observations (*V*, *S*) and calculated from $\beta_F$ (*$V_F$*, *$S_F$*) using the conversion factors in Eq.





(4). Standard deviations of $V_F$ and $S_F$ from corresponding values of $V_{3\beta+2\alpha}$ and $S_{3\beta+2\alpha}$ are 10% and
25% respectively.

The conversion factors in Eq. (4) are now used to estimate the smoke microphysical

properties inside the cloud, assuming in addition that the predominant contribution to the
fluorescence is provided by the smoke. Table 2 summarizes the number, surface area, and
volume concentrations of smoke particles inside the ice clouds, estimated from fluorescence
measurements for four episodes considered in this paper. On September 12-13, 2020, the smoke
layer with high fluorescence and low depolarization ratio at 1064 nm (below 4%) was observed
during the whole night inside the 2.0 km – 5.0 km height range. The cirrus cloud occurred above
11000 m also during the whole night. Fig.6a presents vertical profiles of the aerosol $\beta_{532}$ and
fluorescence $\beta_F$ backscattering coefficients. Fluorescence backscattering shows a maximum at
3.5 km, but it is detected even inside the cloud. The smoke backscattering coefficient $\beta_{532}^s$,
computed from $\beta_F$ for $G_F=3.6\times10^{-4}$ agrees well with $\beta_{532}$ inside the 2.0 – 10.0 km height range
(Fig.6b). The height profile of the surface area concentration of the smoke particles, calculated
from $\beta_F$ using the respective conversion factor in Eq. (4), is shown in Fig.6c. In the smoke layer,
S is up to 60 μm$^2$/cm$^3$, while in the center of the cloud in 12 km – 13 km height the average value
of S is 1.6±0.4 μm$^2$/cm$^3$. Corresponding values of number and volume concentrations in the
cloud center are 3.5±1.8 cm$^{-3}$ and 0.13±0.013 μm$^3$/cm$^3$.

The temperature in the cloud ranged from about -50°C to almost -70°C and was -68°C at

cirrus top in Fig.6b where ice nucleation usually starts. We applied the immersion freezing INP
parameterization of Knopf and Alpert (2013) for Leonardite (a standard humic acid surrogate
material) and assume that this humic compound represents the amorphous organic coating of
smoke particles. The INP parameterization for smoke particles is summarized for lidar
applications in Ansmann et al. (2021). The selected parameterization allows the estimation of the
INP concentration as a function of ambient air temperature (freezing temperature), ice
supersaturation, particle surface area, and time period for which a certain level of ice
supersaturation is given. We simply assume a constant ice supersaturation of around 1.45 during
a time period of 600 s (upwind phase of a typical gravity wave in the upper troposphere). The
temperature at cirrus top height is set to -68°C and the aerosol surface area concentration to 2.0
μm$^2$/cm$^3$ as indicated in Fig.6c. The obtained INP concentrations of 1-10 L$^{-1}$ for these
meteorological and aerosol environmental conditions can be regarded as the predicted number





concentration of ice crystals nucleated in the cirrus top region. Ice crystal number concentration
of 1-10 $L^{-1}$ are typical values in cirrus layers when heterogeneous ice nucleation dominates. It
should be mentioned that the required very high ice supersaturation levels of close to 1.5 (ice
supersaturation of 1.1-1.2 is sufficient in case of mineral dust particles) are still lower than the
threshold supersaturation level of >1.5 at which homogeneous freezing starts to dominate. At
low updraft velocities around 10-25 cm/s, as usually given in gravity waves in the upper
troposphere (Barahona et al., 2017), heterogeneous ice nucleation very likely dominates the ice
production when cirrus evolves in detected aerosol layers.

***3.3. Ice formation inside the smoke layers***.

During September 2020 we observed several episodes with ice cloud formation inside of
smoke layers. One of such episodes occurred on 11-12 September 2020 and is shown in Fig.7.
The height – time display of the fluorescence backscattering coefficient reveals the smoke layer
in the 5.0 - 10.0 km height range. Inside this layer, we can observe a short time interval of 15
minutes with a strongly increased depolarization ratio around 10.5 km height (red spots),
indicating ice cloud formation. Fig.8 shows vertical profiles of the aerosol backscattering
coefficients $\beta_{355}$, $\beta_{532}$, and $\beta_{1064}$ as well the particle depolarization ratios $\delta_{355}$, $\delta_{532}$, and $\delta_{1064}$ for
two temporal intervals. The first interval (23:00 – 00:30 UTC) is prior to ice cloud formation and
the second one (01:20 – 01:45 UTC) covers ice occurrence period. The depolarization ratios at
all three wavelengths were < 5% below 6 km height. Above that height $\delta_{355}$ significantly
increased reaching the value of 10% at 7 km (Fig.8b), which is indicative of a change of the
particle shape (from spherical to irregular shape). The fluorescence capacity also changed with
height, being about $G_F=4.5\times10^{-4}$ at 5.5 km and it decreases to $3.5\times10^{-4}$ by 8 km. The profile of
$\beta_{532}^{s}$ shown in Fig.8c is calculated assuming $G_F=4.0\times10^{-4}$ and it matches well the profile of $\beta_{532}$
for the whole height range. The aerosol layer at 10.5 km is thus a pure smoke layer. Ice
formation at 10.5 km (Fig.8d-f) leads to a significant increase of $\beta_{532}$  while  $\beta_{532}^{s}$ (or the
respective fluorescence backscatter coefficient $\beta_F$) remains low and at the same level as observed
below the cirrus layer, i.e., below 10 km height. The depolarization ratios at all three
wavelengths increases to typical cirrus values around 40%. The temperature at 10.5 km is about -
50 °C, and the surface area concentration of the smoke particles inside the cloud, estimated from





$\beta_F$, is about 10 $\mu m^2/cm^3$ (see Fig.8f, thin blue line). For these temperature and aerosol conditions
we yield smoke INP concentrations of 1-10 $L^{-1}$ for ice supersaturation values even below 1.4
(1.38-1.4) and updraft duration of 600 s. When comparing Fig.8c and 8f at cirrus level it seems
to be that ice nucleation on the smoke particles widely depleted the smoke INP reservoir.

Another case of ice formation in the smoke layer was observed on 17-18 September

2020. Strong smoke layers occurred in the 5.0 km – 9.0 km height range as shown in Fig.9.
During the period from 22:30 – 00:00 UTC, the depolarization increased at 8.5 km height,
indicating ice formation. Vertical profiles of the particle parameters prior and during ice
formation are shown in Fig.10. The $\beta^s_{532}$ calculated for $G_F=3.5\times10^{-4}$ matches well with $\beta_{532}$
below 6.9 km and above 8.0 km (Fig.10c), but inside the 7.0 km – 8.0 km height range
$\beta_{532} > \beta^s_{532}$, meaning that $G_F$ was decreased. The depolarization ratio in the 7.0 km – 8.0 km
height range shows some enhancement (Fig.10b): in particular, $\delta_{532}$ increased from 10% to 12%.
Cloud formation at 8.5 km (Fig.10d) led to a significantly smaller increase of the depolarization
ratio, compared to the case on 11-12 September. Prior to the cloud formation the values of $\delta_{1064}$,
$\delta_{532}$, and $\delta_{355}$ at 8.5 km were of 3%, 10%, and 13% respectively (Fig.10b) and in the cloud
corresponding depolarization ratios increase up to 9%, 15%, 20%. The reason is probably that
the signal averaging period from 22:45 to 23:45 UTC includes cloud-free section. Three gravity
waves obviously crossed the lidar field site and triggered ice nucleation just before 23 UTC, 15-
30 minutes after 23 UTC, and around mid night (00:00 UTC). The temperature at cloud top at
about 8.5-8.6 km height was close to -35°C. For this high temperature and the high particle
surface area concentration of 200 $\mu m^2/cm^3$ (see Fig.10d, thin blue line) we yield smoke INP
concentrations of 1-10 $L^{-1}$ for a relatively low ice supersaturation of 1.30-1.33 and an updraft
period of 600 s. Again, a depletion of the INP reservoir is visible after formation of the cirrus
layer (see Figs.6c and 6d around and above 8.5 km height).

**Conclusion**

The operation of a fluorescence channel in the LILAS lidar during strong smoke events in

the summer and autumn seasons of 2020 has demonstrated the ability of the fluorescence lidar
technique to monitor smoke layers in the UTLS height range in large detail. The fluorescence
capacity $G_F$ of smoke particles during this period varied within a relatively small range: 2.8-





$4.5 \times 10^{-4}$, thus the use of the mean value of $G_F$ allows to estimate the contribution of smoke to the
total particle backscattering coefficient. The fluorescence lidar technique makes it possible to
estimate smoke parameters, such as number, surface area and volume concentration in UTLS
height range in a quantitative way by applying conversion factors ($C_N$, $C_S$, $C_V$) which link the
fluorescence backscattering and the smoke microphysical properties. These factors, derived from
the synergy of multiwavelength Mie-Raman and fluorescence lidar observations, show some
variation from episode to episode, however, the use of mean values of $C_N$, $C_S$, $C_V$ allow
estimation of smoke properties in UTLS height regime with reasonable accuracy. Based on two
case studies, we demonstrated that the fluorescence lidar technique provides the unique
possibility to characterize the smoke particles and their amount inside cirrus cloud layers. The
smoke input parameter (surface area concentration) in smoke INP parameterization schemes that
are used to predict ice crystal number concentrations in cirrus generation cells, can now be
estimated within freshly formed cirrus layers.

The smoke parameters such as fluorescence capacity and conversion factors were derived

from observations of aged wildfire smoke, transported over Atlantic in 2020. However, smoke
composition, depends on many factors, such as burning materials type, flame temperature and
environmental conditions, thus the smoke fluorescence properties may also vary. Hence, it is
important to perform the measurements for different locations and seasons. The fluorescence
backscattering in UTLS height range is quite weak, so to perform measurements with higher
temporal resolution more powerful lidar systems are needed. A dedicated high power Lidar,
LIFE (Laser Induced Fluorescence Explorer), will be designed and operated at ATOLL, in the
frame of OBS4CLIM/ACTRIS-France . .


**Acknowledgement**
We acknowledge funding from the CaPPA project funded by the ANR through the PIA under
contract ANR-11-LABX-0005-01, the "Hauts de France" Regional Council (project CLIMIBIO)
and the European Regional Development Fund (FEDER). The "Réseau National de Surveillance
Aérobiologique" (RNSA) and the "Association pour la Prévention de la Pollution
Atmosphérique" (APPA) are gratefully acknowledged for providing Hirst-collected pollen grains


identification and for assistance with the pollen data handling. Development of algorithm for
analysis of fluorescence observations was supported by Russian Science Foundation (project 21-
17-00114). ESA/QA4EO program is greatly acknowledged for his support to the observation
activity at LOA.

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





Table 1. Conversion factors $C_N$, $C_S$, and $C_V$, and fluorescence capacity $G_F$ at height $H$ for five
smoke episodes. Volume and surface area concentration of smoke particles, obtained by the
inversion of $3\beta+2\alpha$ lidar observations ($V$, $S$), are given together with values calculated from
fluorescence measurements ($V_F$, $S_F$) and using conversion factors (Eq. 4).

| Date dd/mm/yy | $H$, km | $C_N$, $10^4 \frac{cm^{-3}}{Mm^{-1}sr^{-1}}$ | $C_S$, $10^4 \frac{\mu m^2 cm^{-3}}{Mm^{-1}sr^{-1}}$ | $C_V$, $10^4 \frac{\mu m^3 cm^{-3}}{Mm^{-1}sr^{-1}}$ | $G_F$, $10^{-4}$ | $V$, $\mu m^3/cm^3$ | | $S$, $\mu m^2/cm^3$ | |
|---|---|---|---|---|---|---|---|---|---|
| | | | | | | $V$ | $V_F$ | $S$ | $S_F$ |
| 23/06/20 | 4.9 | 88 | 35 | 2.4 | 3.5 | 21 | 19 | 306 | 237 |
| 11/09/20 | 7.5 | 75 | 28 | 2.0 | 3.9 | 7.6 | 8.7 | 111 | 111 |
| 14/09/20 | 6.0 | 90 | 34 | 2.3 | 3.7 | 6.4 | 6.1 | 94 | 78 |
| 17/09/20 | 6.8 | 21 | 21 | 2.3 | 2.9 | 8.0 | 7.8 | 73 | 100 |
| 20/09/20 | 4.9 | 33 | 22 | 2.0 | 4.3 | 2.7 | 2.9 | 31 | 37 |

Table 2. Number $N$, surface area $S$, and volume $V$ concentrations of smoke particles inside the
ice cloud at height $H$ estimated from fluorescence measurements by applying the conversion
factors in Eq. (4) for four measurement sessions.

| Date dd/mm/yy | Time UTC | $H$, km | $\beta_F$, $10^{-4} Mm^{-1}sr^{-1}$ | $N$, $cm^{-3}$ | $S$, $\mu m^2/cm^3$ | $V$, $\mu m^3/cm^3$ |
|---|---|---|---|---|---|---|
| 12/09/20 | 01:20-01:45 | 10.5 | 0.32 | 20±10 | 9±2.3 | 0.7±0.07 |
| 12-13/09/20 | 21:00-03:00 | 12.5 | 0.06 | 3.5±1.8 | 1.6±0.4 | 0.13±0.013 |
| 17/09/20 | 22:45-23:45 | 8.5 | 6.5 | 400±200 | 180±45 | 14±1.4 |
| 24-25/11/20 | 18:00-06:00 | 8.0 | 0.013 | 0.8±0.4 | 0.36±0.09 | 0.03±0.003 |




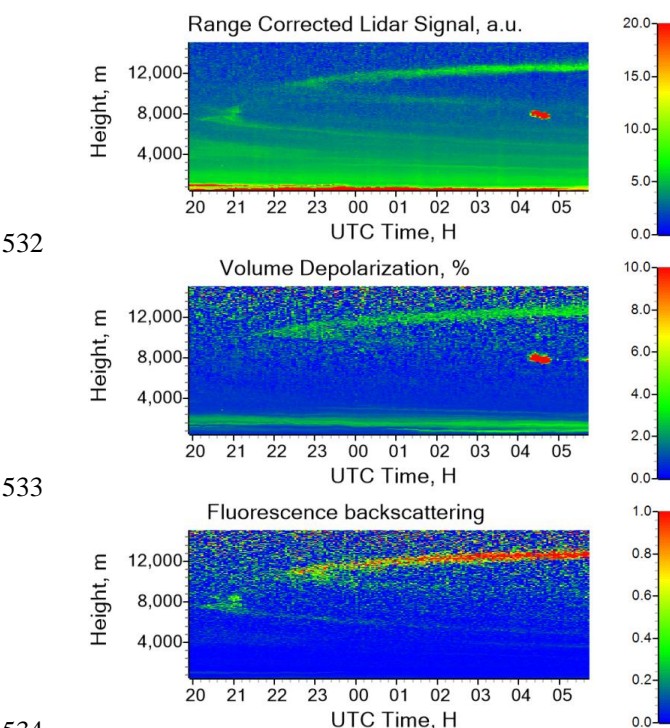


Fig.1. Range corrected lidar signal at 1064 nm, volume depolarization ratio at 1064 nm and
fluorescence backscattering coefficient (in $10^{-4}$ $Mm^{-1}sr^{-1}$) on 4-5 November 2020.





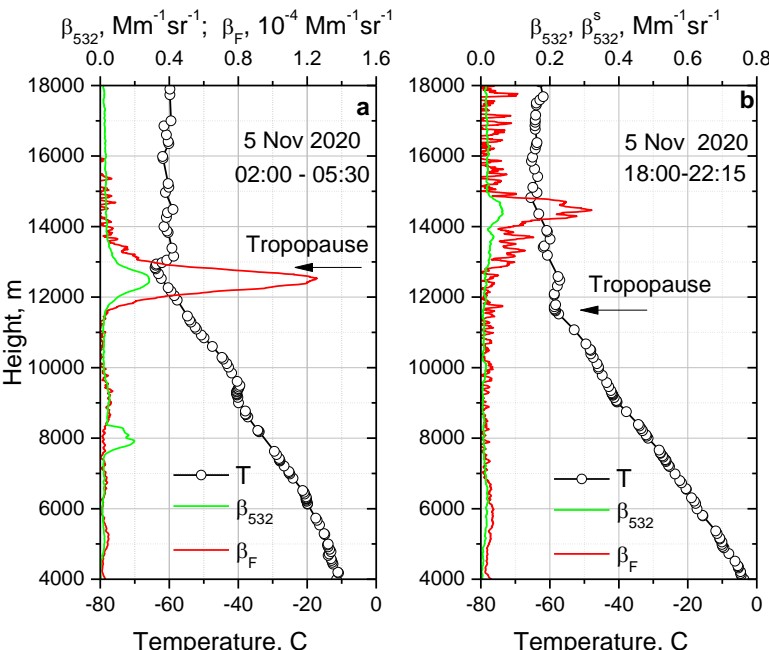

Fig.2. Vertical profiles of aerosol backscattering $\beta_{532}$ and fluorescence backscattering $\beta_F$
coefficients on 5 November 2020 for the periods (a) 02:00 - 5:30 UTC and (b) 18:00 – 22:15
UTC. Open symbols show the temperature profile measured by the radiosonde launched at
Herstmonceux (UK).




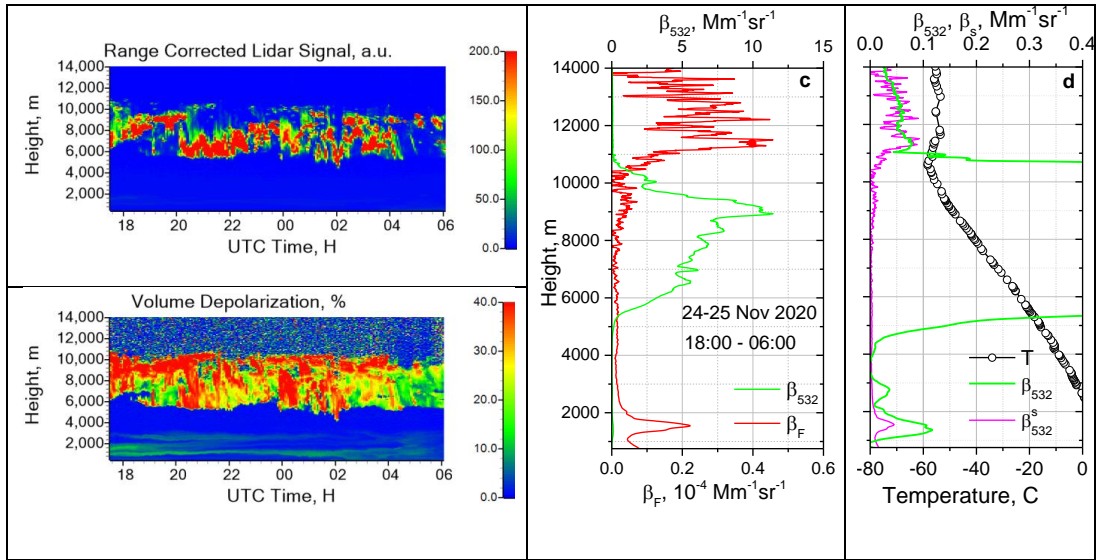


Fig.3. Smoke fluorescence in the presence of clouds on 24 – 25 November 2020. (a, b) Height –
temporal distribution of the range corrected lidar signal and volume depolarization at 1064 nm.
(c) Vertical profiles of the aerosol $\beta_{532}$ and fluorescence $\beta_F$ backscattering coefficients. (d)
Aerosol backscattering $\beta_{532}$ together with smoke backscattering $\beta_{532}^s$ coefficient, computed from
$\beta_F$ for $G_F=4.5\times10^{-4}$. Open symbols show temperature profile measured by the radiosonde at
Herstmonceux.






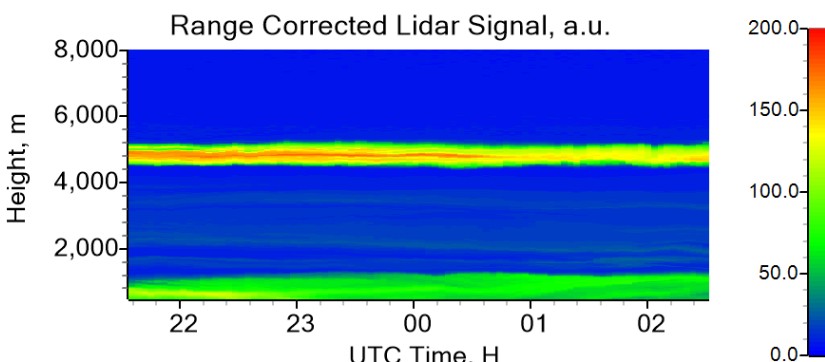

Fig.4. Range corrected lidar signal at 1064 nm on 23-24 June 2020, indicating a thick smoke
layer between 4500 and 5200 m height.

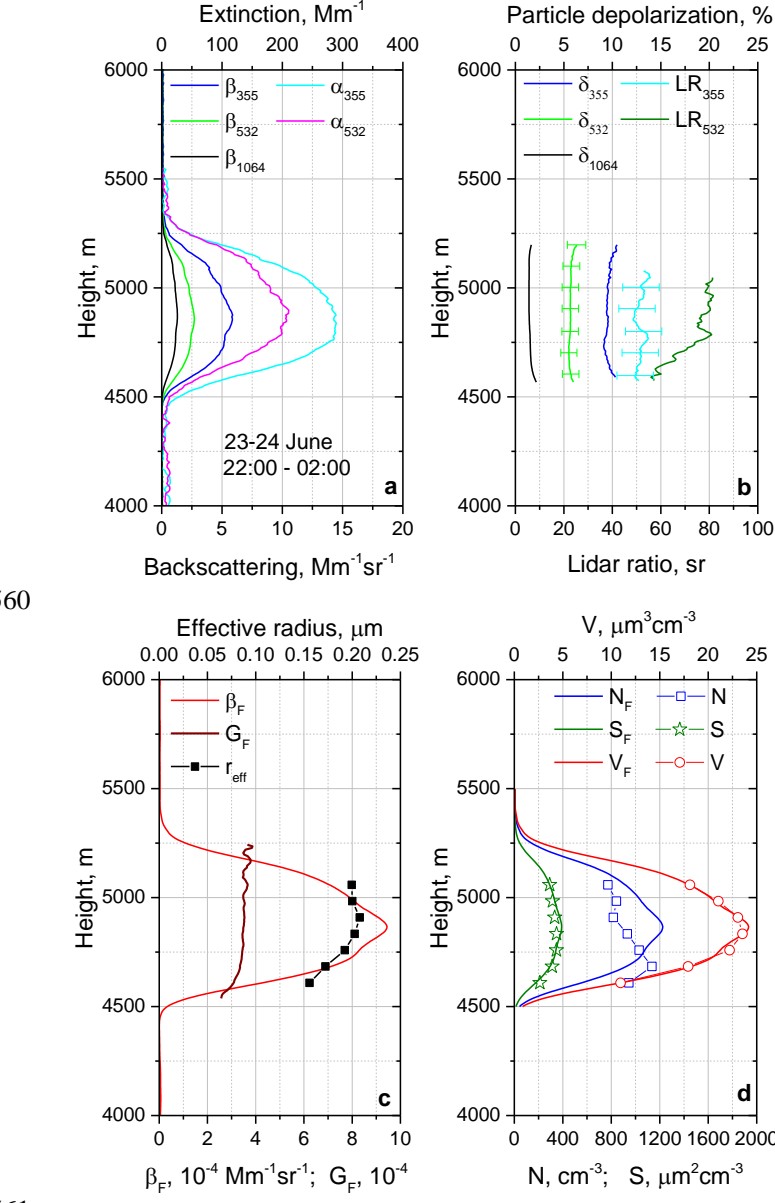


Fig.5. Smoke layer on 23-24 June 2020. (a) Vertical profiles of backscattering ($\beta_{355}$, $\beta_{532}$, $\beta_{1064}$)
and extinction ($\alpha_{355}$, $\alpha_{532}$) coefficients. (b) Particle depolarization ratios ($\delta_{355}$, $\delta_{532}$, $\delta_{1064}$) and lidar
ratios (LR$_{355}$, LR$_{532}$). (c) Fluorescence backscattering ($\beta_F$), fluorescence capacity ($G_F$) and the
particle effective radius (r$_{eff}$). (d) Number ($N$, $N_F$), surface area ($S$, $S_F$) and volume ($V$, $V_F$)
concentrations obtained by inversion of 3$\beta$+2$\alpha$ observations (symbols) and calculated from the
fluorescence backscattering (lines) by using the mean conversion factors defined in Eq. (3).

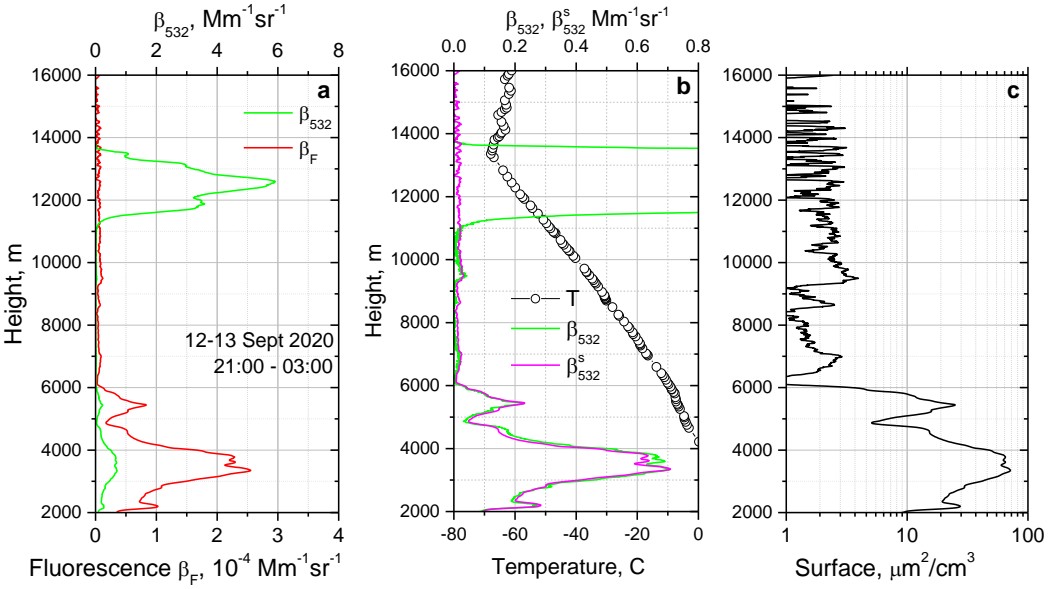

Fig.6. Observation of smoke fluorescence on 12-13 September 2020, 21:00 – 03:00 UTC. (a)
Vertical profiles of the aerosol backscattering $\beta_{532}$ and fluorescence backscattering $\beta_F$
coefficients. (b) Aerosol backscattering $\beta_{532}$ together with smoke backscattering $\beta_{532}^s$ coefficient
computed from $\beta_F$ for $G_F=3.6\times10^{-4}$. (c) Surface area concentration of the smoke particles
calculated from $\beta_F$ by using the respective conversion factor in Eq. (4). Open symbols show the
temperature profile measured by the radiosonde at Herstmonceux.


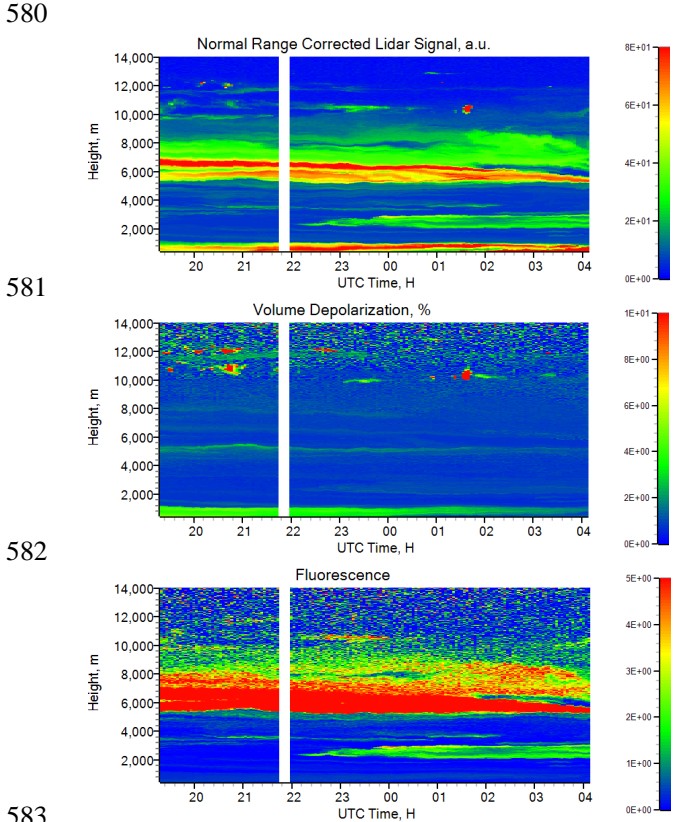



Fig.7. Formation of ice particles at heights above 10 km inside a smoke layer on 11-12 September 2020.
(a) Range corrected lidar signal at 1064 nm, (b) volume depolarization ratio at 1064 nm and (c)
fluorescence backscattering coefficient (in $10^{-4}$ $Mm^{-1}sr^{-1}$).



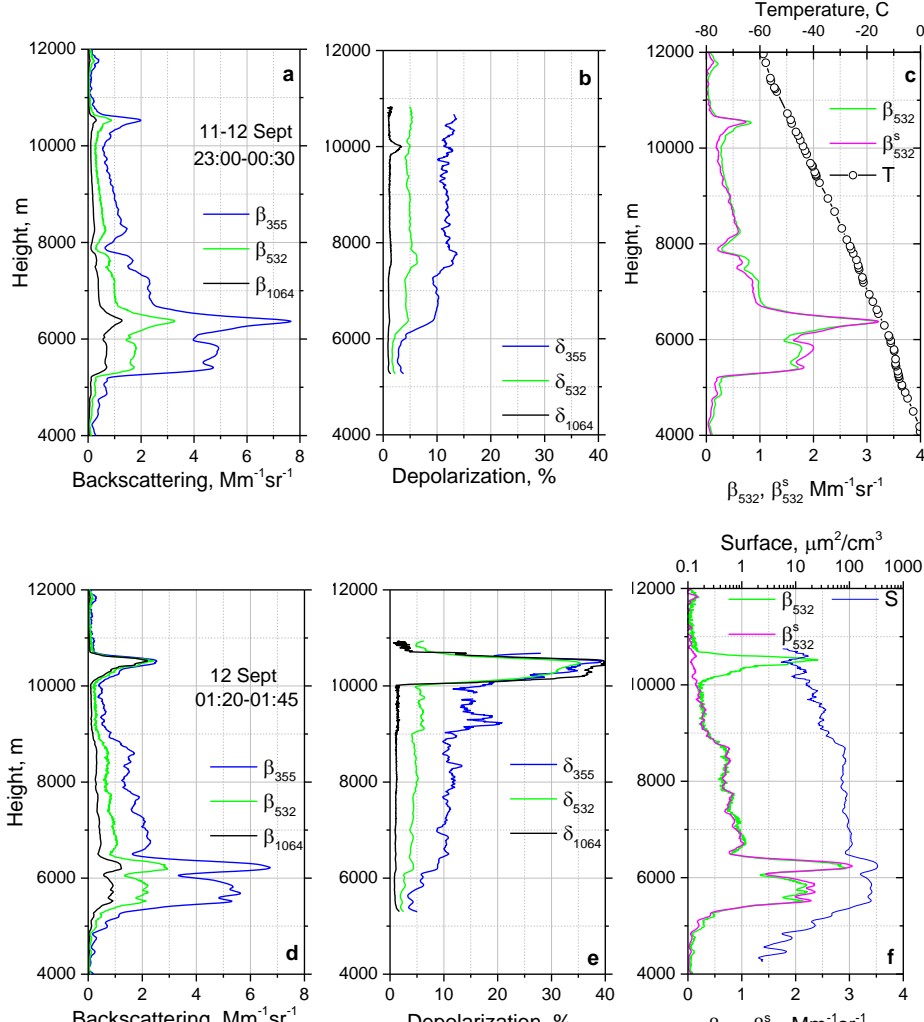


Fig.8. Formation of ice particles at 10-11 km height inside a smoke layer on 11-12 September
2020. Vertical profiles (a, d) of aerosol backscattering coefficients $\beta_{355}$, $\beta_{532}$, $\beta_{1064}$; (b, e) the
particle depolarization ratios $\delta_{355}$, $\delta_{532}$, $\delta_{1064}$; (c, f) backscattering coefficient $\beta_{532}$ together with
backscattering coefficient of smoke $\beta_{532}^{s}$, calculated from fluorescence backscattering $\beta_F$
assuming $G_F=4.0\times10^{-4}$. Plot (f) shows also the surface area concentration $S$ of the smoke
particles calculated from $\beta_F$ by applying the respective conversion factor in Eq. (4). Results are given for
the time intervals 23:00 – 00:30 UTC and 01:20 – 01:45 UTC: prior and during ice cloud
formation at 10.5 km height. The temperature profile measured by the radiosonde at
Herstmonceux is shown with open symbols in panel (c).

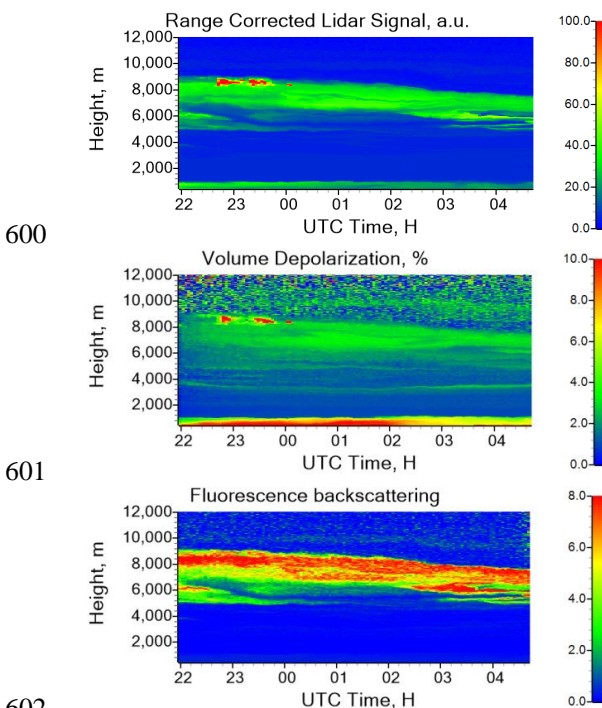

Fig.9. Formation of ice particles at heights above 8 km inside the smoke layer on 17-18 September
2020. (a) Range corrected lidar signal at 1064 nm, (b) volume depolarization ratio at 1064 nm
and (c) fluorescence backscattering coefficient (in $10^{-4}$ $Mm^{-1}sr^{-1}$).


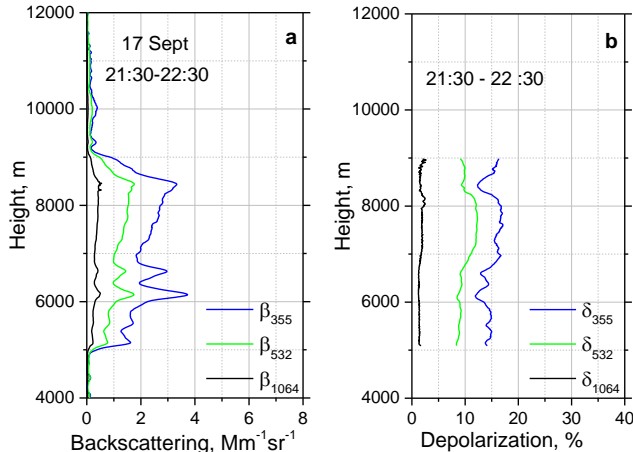


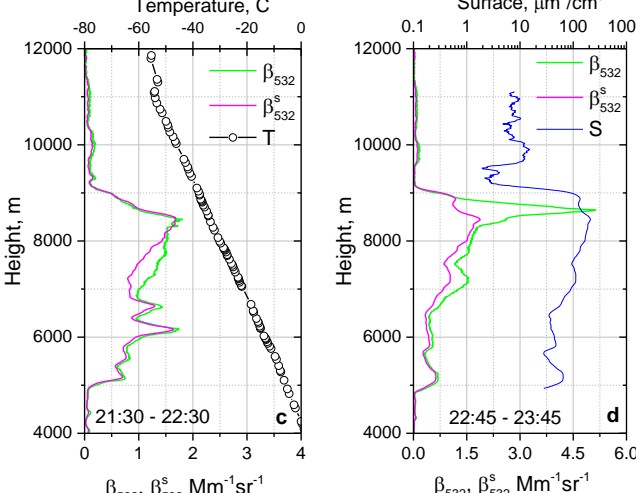

Fig.10. Formation of ice particles at 8.5-8.6 km height inside a smoke layer on 17 September
2020. Vertical profiles of (a) aerosol backscattering coefficients $\beta_{355}$, $\beta_{532}$, $\beta_{1064}$; (b) the particle
depolarization ratios $\delta_{355}$, $\delta_{532}$, $\delta_{1064}$; (c, d) backscattering coefficient $\beta_{532}$ together with
backscattering coefficient of smoke $\beta_{532}^s$, calculated from fluorescence backscattering $\beta_F$
assuming $G_F=3.5\times10^{-4}$. Results are given for the time intervals (a-c) 21:30 – 22:30 UTC and (d)
22:45 – 23:45 UTC: prior and during ice formation at 8.5 km height. Plot (d) shows also the
surface area concentration of the smoke particles calculated from $\beta_F$ and by applying the
respective conversion factor in Eq. (4). The temperature profile measured by the radiosonde at
Herstmonceux is shown with open symbols in panel (c).