# Peer review of "Fluorescence lidar observations of wildfire smoke inside cirrus: A contribution to smoke- cirrus - interaction research"

_Atmospheric Chemistry and Physics, 2021_

## Referee Comment (RC2)

[referee-annotated manuscript omitted]

---

## Author Comment (AC1)

**Reviewer 1**

First of all we would like to thank reviewer for careful reading the manuscript and for appreciation of our work.

*The manuscript "Fluorescence lidar observations of wildfire smoke inside cirrus: A contribution to smoke-cirrus – interaction research" is very well written and presents a novel apprach to combine Flourescence and Multiwavelength Raman Lidar to better isolate vertically profiled smoke characteristics both in the free atmosphere as well as in cirrus clouds. The approach is quite empirical and intersting and should provide a uselfull tool in exploring a number of aerosol - cloud interctions and how smoke can mix and be modified in cloudy backround enironments.*

*One issue is that it is not clear what kind of quantitative errors we are propogating into the Flouresence retreivals (both Flourescence Backscatter and Smoke Microphysical). In particular, estimates of plausible errors are not shown in Tables 1 or 2 whereas the aerosol parameter errors are estimated.*

The question about uncertainty of the method is definitely very important. Uncertainties of $\beta_F$ calculation depend on the chosen value of $\sigma_R$ and on relative transmission of optical elements in fluorescence and nitrogen channels. These system parameters do not change with time. The relative sensitivity of PMTs, however, may change. Regular calibration of the channels demonstrates that corresponding uncertainty can be up to 10%. At high altitudes the statistical uncertainty becomes predominant.

We should recall, that when we estimate smoke concentration from fluorescence backscattering, using mean conversion factors from Eq.(4), the mentioned above systematical uncertainties disappear, because conversion factors and $\beta_F$ are calculated with the same assumptions. The uncertainties of $N, S, V$ estimation arise mainly due to variation of conversion factors for different smoke episodes. We estimate corresponding uncertainties to be below 50%, 25% and 20% for number, surface and volume density respectively.

In revised manuscript we added paragraph describing possible uncertainties of $\beta_F$ calculation. Uncertainties are added to Table 1 and to the plots in the figures.

---

## Author Comment (AC2)

**Reviewer 2**

We would like to thank reviewer for useful suggestions and numerous corrections, which he introduced in our manuscript.

Below we answer his comments

*The paper presents a methodology based on fluorescence lidar measurements, that allows to detect and to quantify the smoke content in upper troposphere and lower stratosphere (UTLS). The methodology is based on several assumptions which are not well validated. My major conern is about the methodology to retrieve the values of $N_F$ $S_F$ $V_F$ (lines 193-199, and 217-219). For instance, the authors explain that they retrieve the N, S, V values from optical data 3β+2α. Then they introduce $C_N$, $C_S$ $C_V$ parameters (eq. 3), based on the previous retrieval of the $β_F$ values. The state that these factors allow the estimation of N, S, V from fluorescence backscatter, although N, S, V values are already known from the optical data 3β+2α. This is a point of confusion.*

To prevent possible confusion, in revised manuscript we use notations *N, S, V* for concentrations derived from 3β+2α observations. For smoke concentration, obtained from fluorescence we use notations $N^S$, $S^S$, $V^S$. Values of *V* and $V^S$ for example, can be close inside smoke layer. But inside cirrus clouds $V^S$ presents just small fraction of *V*.

*Another issue is how they retrieve the $N_F$ $S_F$ $V_F$ values. This is not at all clear in the manuscript. Do these values come from the comparison with the N S V ones from different cases studies?*

To simplify understanding of calculation steps, we followed reviewer suggestion and added in the revised manuscript Appendix with corresponding flow chart.

**Appendix A.** Estimation of smoke parameters from Mie-Raman and fluorescence lidar measurements.

[Figure]

Fig.A. Flow chart showing the main steps of the procedure of smoke parameters estimation from multiwavelength Mie-Raman and fluorescence lidar measurements. Procedure includes the following steps. (i) For a strong smoke layer the $3\beta+2\alpha$ data set, derived from multiwavelength Mie-Raman lidar observations, is inverted to the particle number $N$, surface $S$ and volume $V$ density. (ii) Conversion factors $C_N$, $C_S$, $C_V$ are calculated from Eq.(3) by using the fluorescence backscattering coefficient $\beta_F$. (iii) Different smoke events are analyzed to get mean values of conversion factors $<C_N>$, $<C_S>$, $<C_V>$. These mean values are used to estimate smoke concentration in weak layers in UTLS and inside cirrus clouds in regular observations. The mean value of smoke fluorescence capacity $<G_F>$ allows estimation of smoke contribution $\beta_{532}^s$ to the total backscattering coefficient $\beta_{532}$.

We hope, that now it will help the reader.

*These points need clarification, along with putting error bars in all parameters shown in the various profiles.*

Error bars are added to the plots

*Some minor corrections have to be made, based on the uploaded annotated manuscript. In many places the article "the" is missing. The English text should be revised, as in some places it is unclear.*

We followed reviewer suggestions and introduced modifications in the manuscript.

*PS. I propose to introduce in an appendix or supplement a flow chart showing each calculation step for every retreived parameter [eg. (3β+2α) --> N, S, V; βF --> βS 532 ,etc]. This will facilitate the reader to follow the estimation of the different parameters.*

Yes, Appendix is added

*Ln. 264 "We simply assume a constant ice supersaturation of around 1.45 during a time period of 600 s (upwind phase of a typical gravity wave in the upper troposphere)".*
*Where this assumption is based on? Is there a reference paper to cite?*
Ln. 269. *"Ice crystal number concentration of 1-10 $L^{-1}$ are typical values in cirrus layers when heterogeneous ice nucleation dominates."*
Provide reference papers

It is well accepted that the supersaturation levels for homogeneous ice nucleation need to be about 1.5 to 1.6 to start homogenous freezing. In the presence of INP, nucleation may start at supersaturation level of 1.3 - 1.45 and the supersaturation stops to increase, except the updraft is very strong, which is not the case in the upper troposphere. These mechanisms are discussed in publications listed below. Corresponding references are added to the revised manuscript.

Ansmann, A., Mamouri, R.-E., Bühl, J., Seifert, P., Engelmann, R., Hofer, J., Nisantzi, A., Atkinson, J. D., Kanji, Z. A., Sierau, B., Vrekoussis, M., and Sciare, J.: Ice-nucleating particle versus ice crystal number concentration in altocumulus and cirrus layers embedded in Saharan dust: a closure study, Atmos. Chem. Phys., 19, 15087–15115, https://doi.org/10.5194/acp-19-15087-2019, 2019.

Ansmann, A., Ohneiser, K., Mamouri, R.-E., Knopf, D. A., Veselovskii, I., Baars, H., Engelmann, R., Foth, A., Jimenez, C., Seifert, P., and Barja, B.: Tropospheric and stratospheric wildfire smoke profiling with lidar: mass, surface area, CCN, and INP retrieval, Atmos. Chem. Phys., 21, 9779–9807, https://doi.org/10.5194/acp-21-9779-2021, 2021.

Engelmann, R., Ansmann, A., Ohneiser, K., Griesche, H., Radenz, M., Hofer, J., Althausen, D., Dahlke, S., Maturilli, M., Veselovskii, I., Jimenez, C., Wiesen, R., Baars, H., Bühl, J., Gebauer, H., Haarig, M., Seifert, P., Wandinger, U., and Macke, A.: Wildfire smoke, Arctic haze, and aerosol effects on mixed-phase and cirrus clouds over the North Pole region during MOSAiC: an introduction, Atmos. Chem. Phys., 21, 13397–13423, https://doi.org/10.5194/acp-21-13397-2021, 2021.

Sullivan, S. C., Morales Betancourt, R., Barahona, D., and Nenes, A.: Understanding cirrus ice crystal number variability for different heterogeneous ice nucleation spectra, Atmos. Chem. Phys., 16, 2611–2629, https://doi.org/10.5194/acp-16-2611-2016, 2016.